# Temperature Effect on Ionic Polymers Removal from Aqueous Solutions Using Activated Carbons Obtained from Biomass

**DOI:** 10.3390/ma16010350

**Published:** 2022-12-30

**Authors:** Marlena Gęca, Małgorzata Wiśniewska, Teresa Urban, Piotr Nowicki

**Affiliations:** 1Department of Radiochemistry and Environmental Chemistry, Institute of Chemical Sciences, Faculty of Chemistry, Maria Curie-Sklodowska University in Lublin, M. Curie-Sklodowska Sq. 3, 20-031 Lublin, Poland; 2Department of Applied Chemistry, Faculty of Chemistry, Adam Mickiewicz University in Poznań, Uniwersytetu Poznańskiego 8, 61-614 Poznań, Poland

**Keywords:** activated biocarbons, temperature effect, poly(acrylic acid), polyethyleneimine, polymers simultaneous removal, hydrodynamic radius of polymeric coil

## Abstract

The main aim of this study was the determination of temperature influence on adsorption mechanisms of anionic poly(acrylic acid) (PAA) and cationic polyethylenimine (PEI) on the surface of activated carbons (AC) obtained via chemical activation of nettle (NE) and sage (SA) herbs. All measurements were performed at pH 3 at three temperature values, i.e., 15, 25 and 35 °C. The adsorption/desorption of these polymers from single and mixed solution of adsorbates was also investigated. The viscosity studies were additionally performed to obtain hydrodynamic radius values characterizing polymeric macromolecules conformation in the solution. These data are very important for the explanation of changes of linear dimensions of polymer chains with the rise of temperature caused by the modification of polymer–solvent interactions. Moreover, the XPS studies for the systems showing the highest adsorbed amounts in the specific temperature conditions were carried out. These were the systems containing PEI, PAA and NE–AC activated carbon at 25 °C. In such a case, the maximum adsorption capacity towards PAA macromolecules from a single solution of adsorbate reaches the value of 198.12 mg/g. Additionally, the thermodynamic parameters including the free energies of adsorption, as well as changes in free enthalpy and entropy were calculated.

## 1. Introduction

Carbonaceous materials are widely used in industry and water treatment technologies [1,2]. However, the most popular is their application for heavy metals removal. Therefore, it should not be surprising that heavy metals adsorption on the surface of different types of adsorbents is reported extensively [3,4,5]. On the other hand, many other substances are currently present in wastewater. Thus, the simultaneous removal of molecules of various chemical characteristics is a very important issue, which requires explanation and extensive studies. Adsorption on the surface of carbonaceous adsorbents could also be successfully used for water remediation from all types of pollutants [6,7].

Polymers are present in the wastewaters in great amounts due to their wide application in various branches of industry, agriculture and ecology [8,9,10,11,12,13,14,15,16,17]. They are often synthetic substances, whose effect on organisms have not been completely explained and they can be potentially dangerous. For this reason, new and more effective methods of polymers removal from aquatic environments are constantly sought and aimed at innovative materials of this type that accompany the development of modern technologies. One of the techniques which meets this criterion is adsorption with activated carbons usage [18,19,20]. It should be noted that polymeric chain conformation in the aqueous solution, determining the maximum amounts of adsorbed macromolecules and structure of their surface layer, depends, among others, on the polymer type and its molecular weight, solution pH and ionic strength, as well as temperature [21,22]. All these parameters affect adsorption efficiency and as a consequence of the stability of such systems.

Polymers dilution in the solution proceeds in two stages. In the first one, the polymer swells due to solvent molecule penetration inside the polymer macromolecules, which causes an increase of the polymeric sample volume, maintaining its original shape. In the second stage, the solvent diffuses between polymer macromolecules, separates them from each other and the homogeneous solution is formed [23]. The final polymers conformation in the solution are the result of two component contributions: solvent–segment interactions and segment–segment interactions with van der Walls forces. The solvent whose action on the polymer significantly exceeds the segment–segment interactions is regarded to be the good one. When the polymer–solvent interaction is not able to overcome the van der Waals forces, the polymer precipitates from the solution [24]. The temperature at which the macromolecule size is equal to the undisturbed one is called θ temperature (theta temperature). A solvent at the temperature θ becomes an ideal one. At the temperature lower than θ, polymer–polymer interactions are favorable. The polymer conformation is more coiled in order to minimize polymers–solvent contacts and maximize the polymer–polymer contacts. When the solution temperature is higher than θ temperature, the opposite effect occurs [25]. Polymeric coils develop due to favourable solvent–polymer interactions.

The main aim of the presented manuscript was to determine influence of temperature on an anionic poly(acrylic acid) (PAA) and cationic polyethylenimine (PEI) adsorption mechanism on the surface of activated carbons (AC) obtained from nettle (NE) and the sage (SA) herbs. The simultaneous adsorption/desorption of both polymers with different ionic characteristics was also investigated. The viscosity, adsorption and desorption measurements were performed at three temperature values, i.e., 15, 25 and 35 °C. The viscosity study enabled determination of linear dimensions of polymeric chains in an aqueous solution with the increasing temperature, which can be very helpful in describing the structure of the polymeric layers formed at the activated carbon–solution interface [26]. Understanding the mechanisms controlling the process of adsorption removal of polymers at different temperatures is crucial for the effective control of many technological operations involving them, often carried out in a wide range of temperatures.

## 2. Materials and Methods

### 2.1. Adsorbents and Adsorbates

Activated carbons (AC) used as the adsorbents were obtained from nettle (NE) and sage (SA) herbs. The preparation procedure and physicochemical characteristics were described in detail in the previous manuscript [27]. Both precursors were impregnated with the 50% phosphoric(V) acid solution (STANLAB, Lublin, Poland) and heated in the nitrogen atmosphere (flow rate 200 cm^3^/min) with first step at 200 °C (temperature build-up 5 °C/min) for 30 min and the second one at 500 °C (temperature build-up 5 °C/min) for 30 min. The obtained activated carbons were denoted as NE–AC and SA–AC, respectively. Their textural parameters and acidic/basic properties are presented in Table 1. The textural characterization of the activated carbons was based on the nitrogen adsorption-desorption isotherms measured at –196 °C on a sorptometer ASAP 2020 manufactured by Micrometrics Instrument Corporation (Norcross, GA, USA). The content of the surface functional groups of acidic and basic nature was evaluated according to the Boehm method [28], using volumetric standards of 0.1 mol/dm^3^ NaOH (POCh, Gliwice, Poland) and 0.1 mol/dm^3^ HCl (Chempur, Piekary Śląskie, Poland) as the titrants.

Poly(acrylic acid) (PAA) and polyethylenimine (PEI) were used as adsorbates. Both PAA and PEI were characterized by the weight average molecular weights equal to 2000 Da. Poly(acrylic acid) (Fluka, Saint Louis, USA) is a weak polyelectrolyte with an anionic character (carboxyl groups are present in its macromolecules) with a pK_a_ value of about 4.5. Thus, at pH 3, at which all measurements were carried out (due to maximal adsorption of both polymers under such conditions), its ionization is minimal [29]. Polyethylenimine (Sigma Aldrich, Saint Louis, MO, USA) is a cationic polymer (amine groups are present in its chains) with a pK_b_ value of about 9 [30]. At pH 3, PEI molecules occur in a completely dissociated form. The single (only PAA or only PEI) and mixed (PAA and PEG simultaneously) systems of adsorbates were investigated.

The NaCl solution (POCh, Gliwice, Poland) with a concentration of 0.001 mol/dm^3^ was used as the basic electrolyte. The HCl (Chempur, Piekary Śląskie, Poland) and NaOH (POCh, Gliwice, Poland) solutions with concentrations ranging from 0.01 to 1 mol/dm^3^ were used for the adjustment of the solution pH. All reagents were of analytical grade.

### 2.2. Viscosity Measurements

Viscosity measurements were applied to determine the linear dimensions of the polymer chain in aqueous solution. Based on the Flory–Fox theory, the root-mean-square chain end-to-end distance—${\left(\bar{r^{2}}\right)}^{1/2}$ and the hydrodynamic radius of the polymer coil—*R_h_* were calculated [24,31]. These experiments were performed using the rotational CVO 50 rheometer with the “double gap” measuring system (Bohlin Instruments, Malvern, UK). For this purpose, the following equations were applied:(1)$$R_{h}=f\frac{{\left(\bar{r^{2}}\right)}^{1/2}}{6^{1/2}}$$
(2)$${\left(\bar{r^{2}}\right)}^{1/2}=\sqrt[{3}]{\left(\frac{\left[\eta \right]M}{F}\right)}$$
(3)$$\left[\eta \right]=\underset{c\to 0}{\lim}\eta_{r}$$
(4)$$\eta_{r}=\frac{\frac{\eta_{p}}{\eta_{0}}-1}{c}$$
where: [*η*]—the intrinsic viscosity (determined from dependence of reduced viscosity (*η_r_*) versus the polymer concentration (*c*) by extrapolating the straight line to *c* = 0); *η_p_*—the polymer solution viscosity; *η*_0_—the water viscosity; *F*—the universal Flory–Fox constant approximately equal to 2.1⋅10^21^ (for the polymers in good solvents) [24,31]; *M*—the polymer molecular weight; *f*—the constant value, irrespective of the polymer molecular weight (*f* = 0.66 [32,33]).

Viscosity studies were performed in the supporting electrolyte solution—NaCl with a concentration of 0.001 mol/dm^3^ at pH 3, at which the highest adsorptions of both polymers were observed [27] and at three values of temperatures, namely 15, 25 and 35 °C. The examined concentrations of aqueous polymeric samples were as follows: 10, 50, 100, 150, 200, 300 and 400 ppm.

### 2.3. Adsorption–Desorption Measurements

For adsorption–desorption studies, 10 cm^3^ of suspensions containing 0.001 mol/dm^3^ of NaCl (supporting electrolyte), 200 ppm of appropriate polymer and 0.01 g of activated carbon were used. The adsorption process was carried out for 24 h at pH 3 at 15, 25 and 35 °C. After the adsorption completion, the solids were separated from solutions using microcentrifuge (Centrifuge MPW 233e MPW Med. Instruments, Warsaw, Poland) and the concentrations of adsorbates in the supernatants were determined. The separated solids with adsorbed macromolecular substances were next subjected to the desorption using H_2_O, HNO_3_ (POCh, Gliwice, Poland) and NaOH solutions (the acid and base with concentrations of 0.1 mol/dm^3^). After 24 h, the adsorbates concentrations were determined. All adsorbed/desorbed amounts were examined using a static method based on a decrease/increase in the polymers concentration in the solution before and after the adsorption/desorption process. The following formula was used for this purpose:(5)$$q_{e}=\frac{\left(C_{0}-C_{e}\right)V}{m}$$
where: *q_e_*—the polymer adsorbed/desorbed amount at the equilibrium state, *C*_0_ and *C_e_*—the polymer concentrations in the solution before adsorption and at the equilibrium state, respectively, *V*—the volume of the solution, *m*—the mass of the solid.

PEI and PAA concentrations in the solution were determined using UV-VIS spectrophotometry (Carry 100, Varian, Palo Alto, CA, USA). The poly(acrylic acid) concentration was determine using its reaction with hyamine 1622 (Sigma Aldrich, Saint Louis, MO, USA), which gives a white-coloured complex [34]. In turn, the polyethylenimine concentration was obtained based on its reaction with CuCl_2_ (POCh, Gliwice, Poland), resulting in the formation of a blue-coloured complex [35].

The PAA and PEI adsorption kinetics were studied at pH 3 at 15, 25 and 35 °C for the NE–AC sample (the time intervals were: 10, 30, 60, 90, 120, 180 min). The procedures of concentration determination described above, as well as the adsorbates with the initial concentration of 100 ppm, were examined. The obtained data were fitted to the pseudo-first order and the pseudo-second order models of adsorption [36]:(6)$$\frac{dq_{t}}{dt}=k_{1}\left(q_{e}-q_{t}\right)$$
(7)$$\frac{dq_{t}}{dt}=k_{2}{\left(q_{e}-q_{t}\right)}^{2}$$
where: *q_e_*—the adsorbed amount at the equilibrium state, *q_t_*—the adsorbed amount after time *t*, *k*_1_—the equilibrium rate constant, *k*_2_—the equilibrium rate constant.

Two different commercially available activated carbon materials were used—purolite (PRT) (Lenntech, Delfgauw, The Netherlands), with a specific surface area of 900–1000 m^2^/g, and lewatite (LWT) (Lanxess, Dortmund, Germany), characterized by a specific surface area of 550–650 m^2^/g. The adsorption studies with their usage were performed at the same conditions as for the applied herbs-based activated carbons.

The XPS (X-ray photoelectron spectroscopy) apparatus (Gammadata Scienta, Uppsala, Sweden) was used to determine elemental composition of NE–AC activated carbon and adsorbed forms in the surface layers of both polymers at pH 3 and at 25 °C. For such a system and under such conditions, the greatest adsorption of polymeric substances was obtained.

### 2.4. Thermodynamic Studies

The thermodynamic parameters of PAA and PEI adsorption on activated carbons were calculated at three different temperatures using the formulas [37,38,39]:(8)$$\Delta G{}^{\circ}=-RTlnK_{c}C_{0}$$
(9)$$K_{c}=\frac{q_{e}}{C_{e}}$$
(10)$$lnK_{c}c^{0}=\frac{\Delta S{}^{\circ}}{R}-\frac{\Delta H{}^{\circ}}{RT}$$
where: Δ*G*°—the change of free energy of the system, Δ*H*°—the change of free enthalpy of the system, Δ*S*°—the change of free entropy of the system, *R*—gaseous constant (8.314 J/mol K), *T*—temperature, *K_c_*—distribution constant at equilibrium, C_0_—the initial concentration of adsorbate (200 mg/dm^3^) [38,39,40], *q_e_*—the adsorbed amount of the polymer at equilibrium, *C*_e_—the equilibrium concentration of solution.

## 3. Results

### 3.1. Linear Dimensions of Poly(Acrylic Acid) and Polyethyleneimine Chains in Aqueous Solutions at Different Temperatures

The dependencies of intrinsic viscosities of polymer solutions as a function of their concentrations leading to [*η*] determination are presented in Figure 1, whereas the calculated parameters characterizing linear dimensions of polymeric chains are listed in Table 2 [40,41].

Analysis of the data above suggests that a temperature increase causes increase of both the root-mean-square chain end-to-end distance and the hydrodynamic radius of the polymer coil values in the case of the two examined polymers. Such behavior is a result of the changes in solvent quality and polymer–solvent interactions [23].

The theta temperature of poly(acrylic acid) in aqueous solution is 14 °C [42]. The dimensions of polymer chains in this state satisfy the statistical requirements of the Gauss theory, which is why they are sometimes termed as the Gauss coils. Under temperature conditions lower than theta temperature, polymer coil conformation changes to a globular shape to minimize the polymer–solvent contacts and maximize the contacts between the segments (coil-globule transition and phases separation occur). PAA aqueous solution shows the UCST type phase diagram (with upper critical solution temperature), below which the solution separates into two uniform phases with different compositions (cloud point is reached) [43]. At 15 °C (very close to the theta temperature), the polymer coils (additionally practically undissociated at pH 3) have the smallest dimensions (0.53 nm). Further increase of temperature causes improvement of solvent quality and polymer segments interactions with solvent molecules more and more preferred. This is why the ${\left(\bar{r^{2}}\right)}^{1/2}$ and *R_h_* values increase at temperatures of 25 and 35 °C.

A similar trend is observed for the cationic polyethyleneimine. Nevertheless, due to the total dissociation of its functional groups at pH 3, the linear dimensions of PEI macromolecules are greater than in the case of PAA. The aqueous solution of PEI shows LCST (lower critical solution temperature) behavior and the reported cloud point is about 97 °C [44,45]. Due to the fact that the examined temperature range (15–35 °C) remains distant from this PEI critical conditions, the differences of the ${\left(\bar{r^{2}}\right)}^{1/2}$ and *R_h_* values with temperature increase are slightly smaller than that observed for poly(acrylic acid) samples.

### 3.2. Adsorption–Desorption Properties of the Activated Carbons in Single and Mixed Solutions of Poly(Acrylic Acid) and Polethyleneimine at Different Temperatures

Both polymers adsorbed amounts (expressed in mg/g and as percentage of polymer removal) on the activated carbons surface from single and mixed systems of adsorbates at pH 3 at three examined temperatures, as presented in Figure 2 and Figure 3.

Previous studies on activated carbons surface charge density confirm the acidic properties of both adsorbents. The point of zero charge (pzc) of NE–AC activated carbon occurs at pH 3.1 and, in case of SA–AC one, at pH 4.0 [27]. It means that at pH 3, at which the polymers adsorption were carried out, NE–AC activated carbon surface is practically neutral, whereas SA–AC one slightly positively charged. Thus, the electrostatic conditions have small influence on the polymer adsorption at pH 3, only for minimally dissociated PAA macromolecules the weak electrostatic attraction with SA–AC surface takes place. Due to the considerably higher content of functional groups and noticeably greater mean pore size in the case of the NE–AC activated carbon (Table 1), PAA adsorption is higher on this carbonaceous material surface than on the SA–AC one, both from single and mixed adsorbate systems, at all examined temperatures. The PEI adsorption is considerably lower as a result of total dissociation of its functional groups and more stretched conformation causing solid pore entry blockade.

The temperature influence on the adsorbed amount of the examined polymers is significantly greater for poly(acrylic acid) (Figure 2) than polyethyleneimine (Figure 3). The largest differences occur for NE–AC activated carbon, for which the adsorption at 15 °C is lower than at 25 and 35 °C, noting that adsorption reaches its maximum at 25 °C (about 198 mg/g). At 15 °C, PAA macromolecules possess nearly ideal conformation (the hydrodynamic radius is the smallest—0.53 nm). Such coiled conformation limited exposition of polymeric functional groups towards the solid surface (which interact each other forming mainly hydrogen bonds [40]) and the smallest PAA adsorption level is observed. Moreover, the temperature conditions close to theta and critical ones make polymer coils less soluble in water, which affects their tendency to accumulate at the interface. The greatest adsorption of PAA on the NE–AC surface at 25 °C is probably a result of the smaller packing of polymeric segments in adsorbed coils with larger size (*R_h_* assumes value 0.77 nm) and the increasing affinity of the polymer to the solvent molecules, the quality of which improves. These factors enable effective interaction of polymer carboxylic groups with the surface of the NE–AC adsorbent and effective penetration of macromolecules into this solid pores. On the other hand, at 35 °C, the PAA adsorption is smaller by about 6 mg/g than at 25 °C, due to further increase of the hydrodynamic radius of the polymeric chains, which results in a more flat conformation of adsorbed PAA chains on the positively charged NE–AC surface. This results in partial blockade of the entrance to the solid pores for adsorbing macromolecules. The temperature influence in the case of PEI and mixed adsorbate systems is rather minimal and, in most cases, it remains within the measurement error.

Both examined polymers show a higher affinity to the surface of activated carbons obtained from plant material than to the surface of the commercially available activated carbons (Figure 4 and Figure 5). Moreover, the effects on mutual adsorption on the surface of LWT and PRT are opposite to that taking place during the adsorption onto NE–AC and SA–AC activated carbons. Poly(acrylic acid) is better adsorbed from the binary solutions on the surface of commercial adsorbents, whereas polyethylenimine adsorbed amounts decrease in the presence of PAA.

The kinetic studies (Figure 6) performed in the NE–AC system as a function of temperature show that the equilibrium state was reached after 60 min in the solution containing poly(acrylic acid) and after 10 min in the solution containing polyethylenimine. The obtained results of kinetic measurements were fitted to the pseudo-first order and pseudo-second order equations (Table 3). The analysis of kinetic parameters indicated that for both polymers, the pseudo-second order model described better their adsorption behavior at the activated carbon–solution interface.

The results of the desorption studies from single and mixed systems of the examined polymers are presented in Table 4. In the case of PAA containing systems, the greatest percentages of desorption were obtained after usage of NaOH (45.5–61.23%). The HNO_3_ turned out to be not very effective in the regeneration of activated carbons, similarly to H_2_O. The desorption of PEI was only realized by the use of water. In this case, desorption percentages of polyethyleneimine from single systems reaches an even value of 44.15%. In the case of mixed systems of adsorbates, the PEI desorption is considerably limited due to the formation of PAA + PEI complexes [27]. This indicates that PEI bonding to the surface of examined carbon materials is much weaker than PAA macromolecules. The influence of temperature on the adsorbed polymers desorption is rather minimal.

The free energy of adsorption (∆*G*°) was calculated using Equations (8) and (9), whereas entropy and enthalpy were evaluated from the van’t Hoff plots presented in Figure 7. All thermodynamic parameters are collected in Table 5.

The negative values of ∆*G°* at all examined temperatures show that the adsorption process of polymeric substances was spontaneous. The greatest values of ∆*G°* were observed in the case of the NE–AC + PAA system at 25 °C, for which the largest adsorption was obtained. The free energy of adsorption changes in the range 9.7–24.7 kJ/mol, which corresponds with the energy of hydrogen bonds formation and electrostatic interactions. The positive values of ∆*H°* confirmed the endothermic nature of the process. Moreover, the positive ∆*S°* values indicate an irregular increase of randomness at the activated carbon–polymer interface during the adsorption process.

### 3.3. XPS Study of Polymeric Adsorption Layers Formed on the NA–AC Activated Carbons Surface at 25 °C

The XPS spectra of the NE–AC sample obtained before and after PAA and PEI adsorption (Figure 8, Figure 9 and Figure 10 and Table 6) confirm the formation of a polymer film on the surface of the tested activated carbon. As a result of poly(acrylic acid) adsorption, a significant decrease in carbon content is observed, accompanied by an increase in the contribution of oxygen. This proves that the polymer (rich in carboxyl groups) is permanently bonded to the adsorbent surface. In the case of the system containing NE–AC activated carbon and polyethyleneimine, a slightly different situation is observed. As a consequence of adsorption of the polymer rich in nitrogen functional species, the contribution of the nitrogen and oxygen increases simultaneously; however, the nitrogen content is almost three times higher than in the system without the polymer. This clearly indicates the permanent bonding of the polymer with the activated carbon surface. We have quite a similar relationship in case of the system containing carbonaceous adsorbent and both polymers at the same time. The increase in the content of both heteroatoms is even greater than for the previously discussed system. This may indicate mutual interactions between the molecules of individual polymers.

According to the data presented in Table 6, as the result of both polymers adsorption, there are not only quantitative changes in the elemental carbon content at the surface, but also qualitative ones. This is especially well seen in the case of the simultaneous removal of PAA and PEI from an aqueous solution. The NE–AC sample contains carbon mainly in the form of such groups as: C-H, C=C, C-C, C-OH and C-O-C, as well as small amounts of C=O, -C=N, N=C-O and COOR. After both polymers adsorption, the contribution of C=C (284.3 eV) and, in particular, C-H (284.8 eV) [46,47,48,49] significantly decreases, while the share of the other forms increases to different degrees. In case of the systems NE–AC + PEI and NE–AC + PEI +PAA, particularly noteworthy is the two-fold increase in the content of C=N, N=C-O (288.8 eV) [50,51,52] and COOR type groups (289.6 eV) [53]. In order to better illustrate the changes taking place on the activated carbon surface as a result of both polymers adsorption, the selected spectra of C 1s, N 1s and O 1s are presented in Figure 9 and Figure 10.

**Table 6 materials-16-00350-t006:** The main C 1s photopeaks of the investigated samples [47,48,49,50,51,52,53].

Species	Binding Energy [eV]	NE_AC	NE_AC + PAA	NE_AC + PEI	NE_AC + PAA + PEI
C-H	284.85	53.2	50.5	47.2	39.9
C=C sp^2^	284.29	12.0	13.5	14.4	9.9
C-C sp^3^	285.51	14.5	15.0	14.0	19.6
C-OH, C-N	286.15	8.2	6.9	5.8	9.2
C-O-C	286.94	6.4	8.3	8.4	10.0
C=O	287.82	1.7	2.1	1.8	3.0
COO-, C=N, N=C-O	288.79	1.9	2.0	4.3	4.6
COOR	289.59	2.2	1.7	4.2	3.8

## 4. Conclusions

The examined carbonaceous materials obtained by chemical activation of nettle and sage herbs can be used as the effective adsorbents of poly(acrylic acid) and polyethyleneimine from the aqueous solutions at different temperatures (changing in the range 15–35 °C). The temperature increase causes an increase in the linear dimensions of polymeric chains (expressed as root-mean-square chain end-to-end distance and the hydrodynamic radius) for both examined polymers. At 15 °C (very close to the PAA theta temperature), the poly(acrylic acid) coils (additionally minimally dissociated at pH 3) have the smallest dimensions (hydrodynamic radius is 0.53 nm).

Due to the total dissociation of polyethylene functional groups at pH 3, the linear dimensions of PEI macromolecules are greater than in the case of PAA. The temperature influence on the adsorbed amounts of the examined polymers is significantly greater for poly(acrylic acid) than polyethyleneimine. The largest differences occur for NE–AC activated carbon, for which the adsorption reaches its maximum at 25 °C (about 198 mg/g). This can be result of the smaller packing of polymeric segments in adsorbed coils and the increasing affinity of the polymer to the solvent molecules (facilitated penetration of macromolecules into solid pores). The thermodynamic studies indicate that the adsorption process of polymeric substances was spontaneous, has endothermic nature, and takes place mainly through hydrogen bonds formation.

It was also proved that poly(acrylic acid) is most efficiently desorbed by the use of a NaOH solution (maximal desorption about 61%). Weakly bonded polyethyleneimine undergo desorption even by water (maximal desorption about 44%). XPS studies confirmed that both polymers form stable films on the surface of the tested activated carbon, leading to significant changes in the content of carbon, oxygen and nitrogen on the border of the adsorbent–adsorbate phases.

## Figures and Tables

**Figure 1 materials-16-00350-f001:**
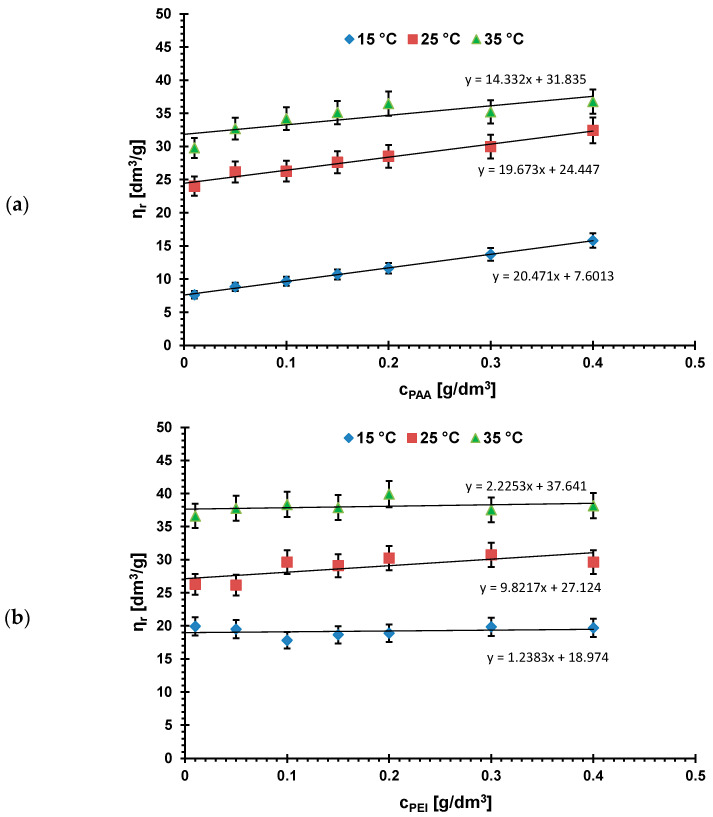
Intrinsic viscosities of (**a**) PAA and (**b**) PEI solutions as a function of their concentrations at different temperatures, pH 3.

**Figure 2 materials-16-00350-f002:**
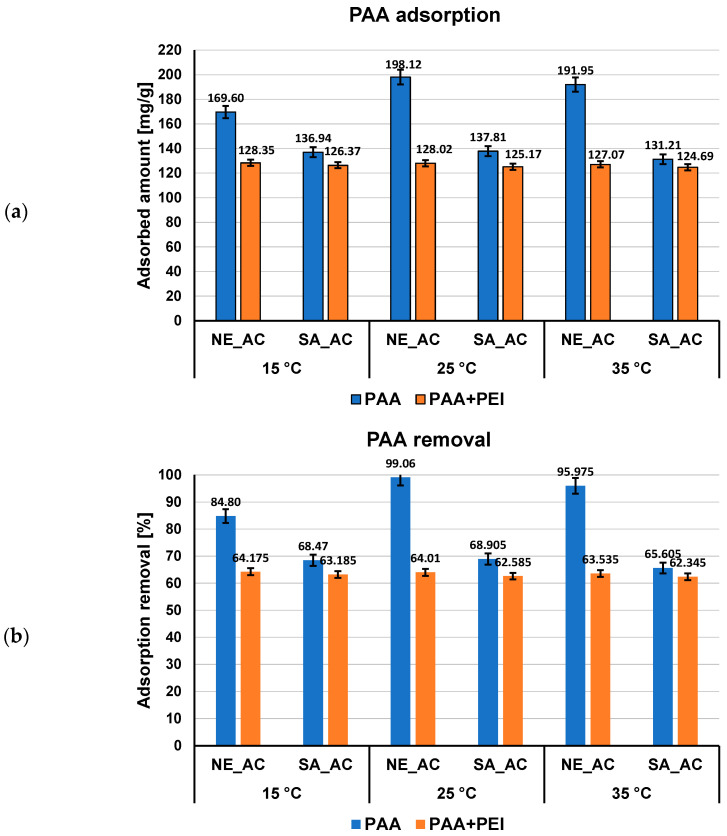
PAA adsorbed amounts (**a**) and its percentage removal (**b**) from the activated carbons surface in single and mixed systems of adsorbates at three examined temperatures, pH 3.

**Figure 3 materials-16-00350-f003:**
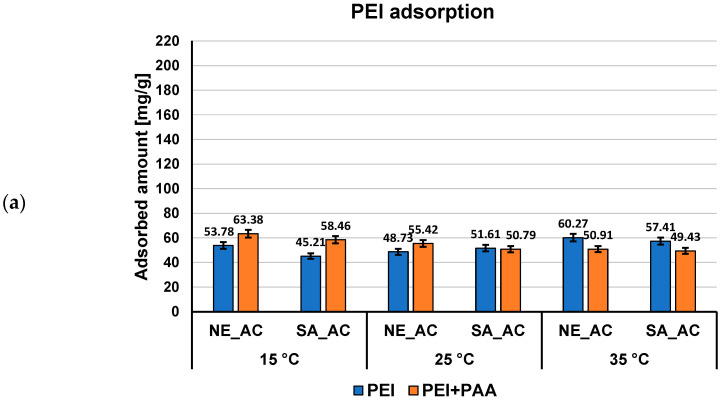
PEI adsorbed amounts (**a**) and its percentage removal (**b**) from the activated carbons surface in single and mixed systems of adsorbates at three examined temperatures, pH 3.

**Figure 4 materials-16-00350-f004:**
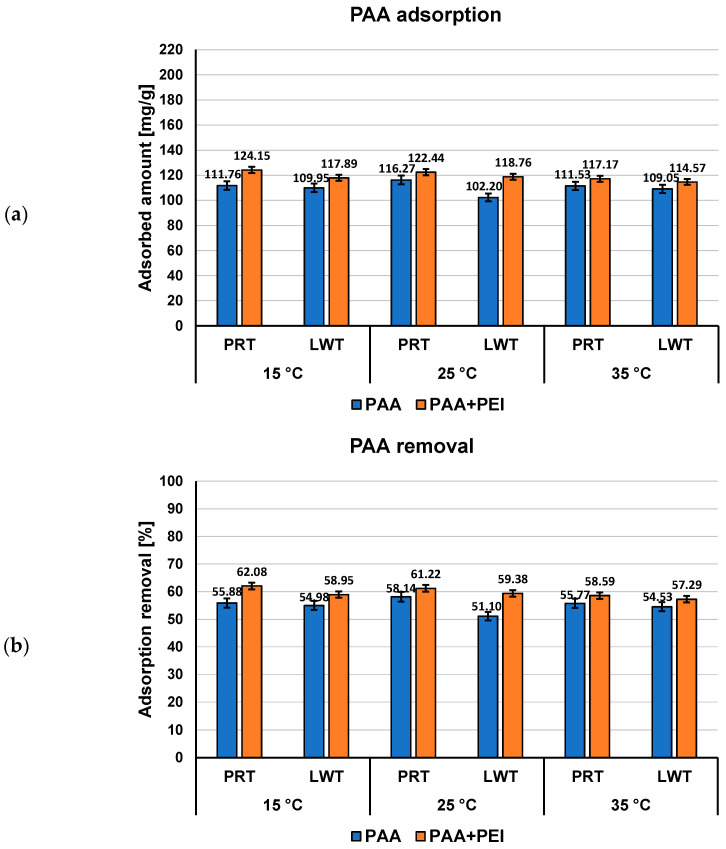
PAA adsorbed amounts (**a**) and its percentage removal (**b**) from the commercial materials surface in single and mixed systems of adsorbates at three examined temperatures, pH 3.

**Figure 5 materials-16-00350-f005:**
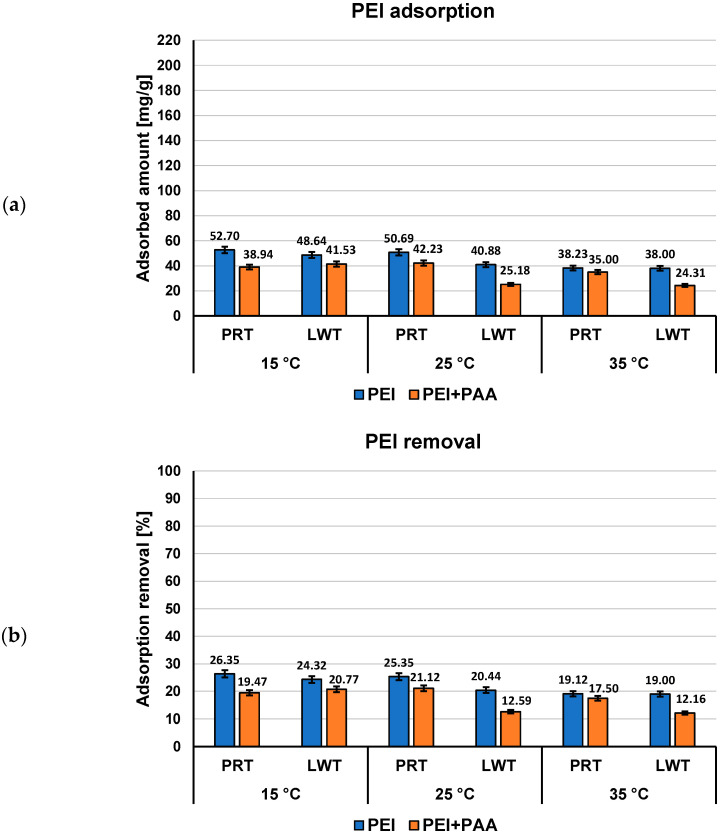
PEI adsorbed amounts (**a**) and its percentage removal (**b**) from the commercial materials surface in single and mixed systems of adsorbates at three examined temperatures, pH 3.

**Figure 6 materials-16-00350-f006:**
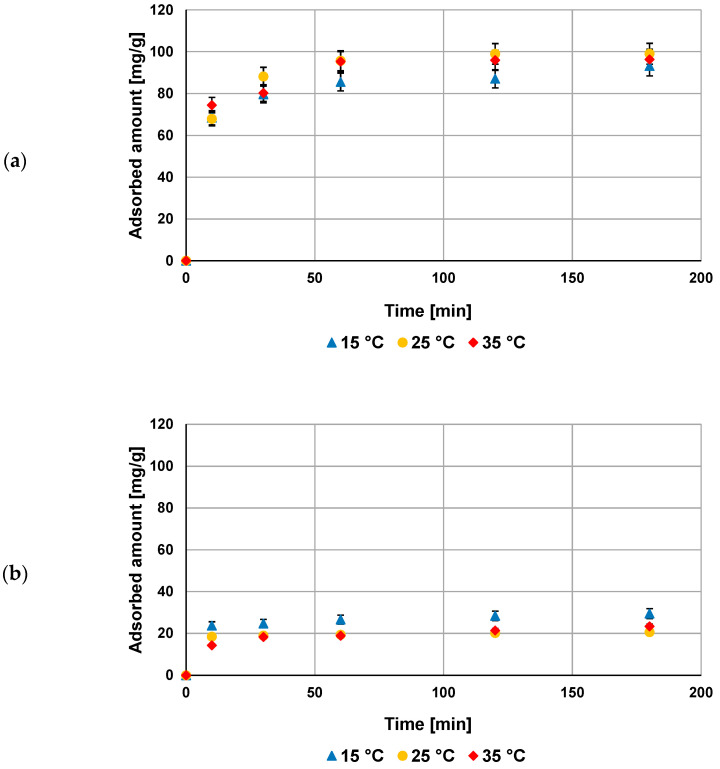
Adsorption kinetics of PAA (**a**) and PEI (**b**) on NE–AC activated carbon surface at three examined temperatures, pH 3.

**Figure 7 materials-16-00350-f007:**
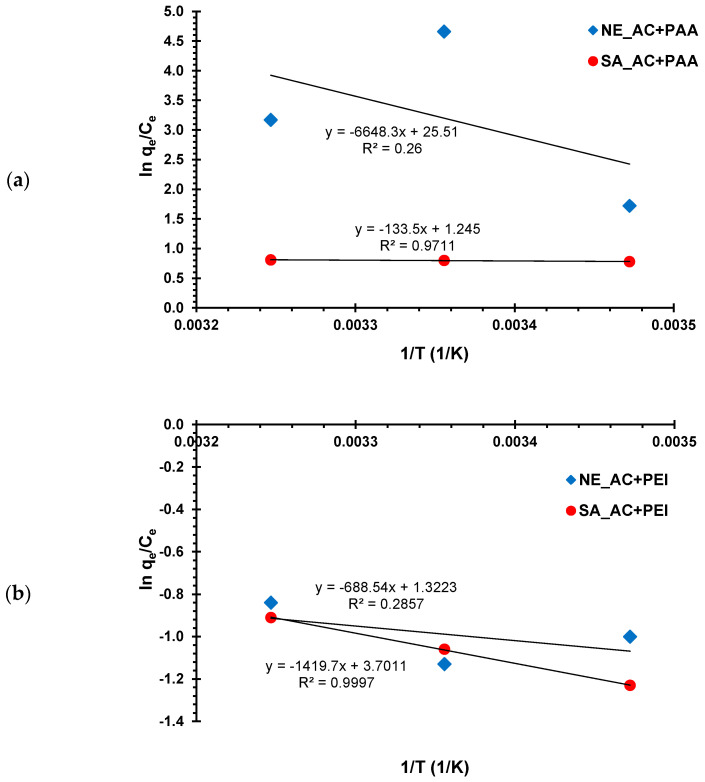
The van’t Hoff plots for calculation of enthalpy and entropy in (**a**) PAA and (**b**) PEI—activated carbon systems.

**Figure 8 materials-16-00350-f008:**
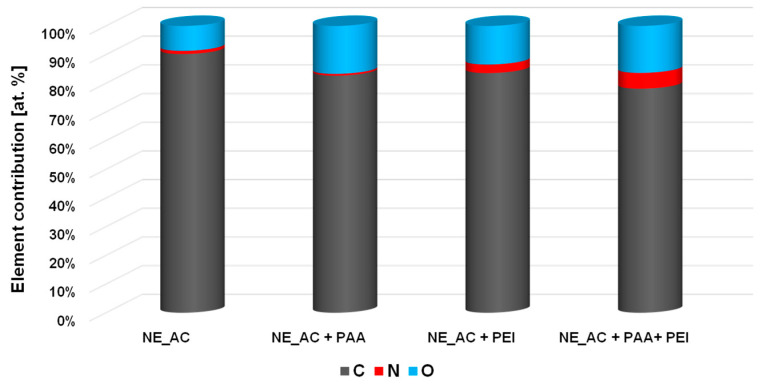
Elemental composition of the activated carbon’s surface before and after adsorption of polymers at 25 °C [at. %].

**Figure 9 materials-16-00350-f009:**
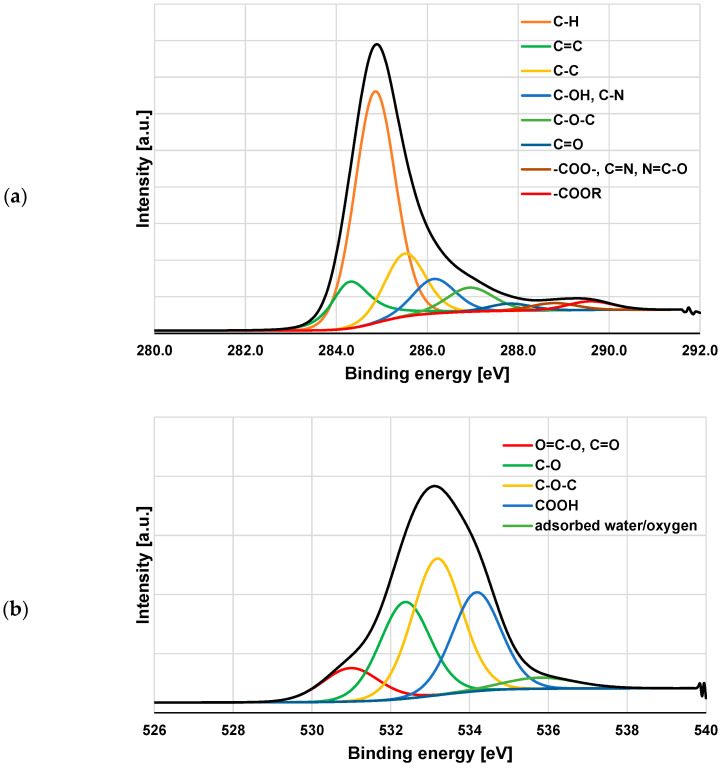
C 1s (**a**) and O 1s (**b**) spectra of NE–AC activated carbon before polymers adsorption.

**Figure 10 materials-16-00350-f010:**
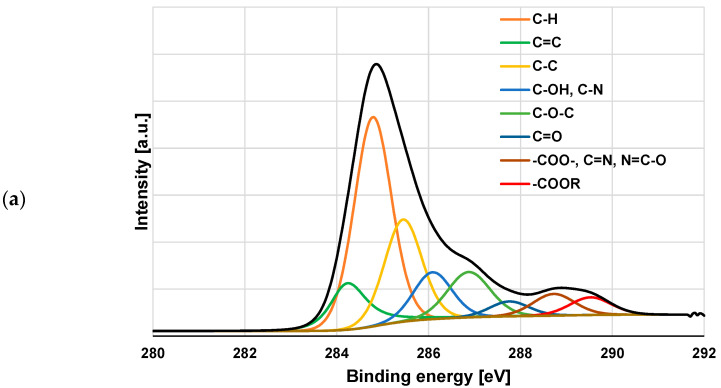
C 1s (**a**), N 1s (**b**) and O 1s (**c**) spectra of NE–AC activated carbon after PAA and PEI simultaneous adsorption.

**Table 1 materials-16-00350-t001:** Textural parameters and acidic/basic properties of the prepared activated carbons.

Adsorbent	Surface Area [m^2^/g]		Pore Volume [cm^3^/g]		Mean Pore Size [nm]	Acidic Groups [mmol/g]	Basic Groups [mmol/g]	Total Amount [mmol/g]
	Total	Micropore	Total	Micropore				
NE–AC	801	157	0.847	0.074	4.231	0.858	0.272	1.130
SA–AC	842	155	0.826	0.074	3.926	0.436	0.215	0.651

**Table 2 materials-16-00350-t002:** Characteristics of linear dimensions of poly(acrylic acid) and polyethyleneimine macromolecules in the aqueous solutions at pH 3 under different temperature conditions.

Temperature	[*η*] [dm^3^/g]	${\left(\bar{r^{2}}\right)}^{1/2}$ [nm]	*R_h_* [nm]
PAA			
15 °C	7.601	1.96	0.53
25 °C	24.447	2.85	0.77
35 °C	31.835	3.16	0.85
PEI			
15 °C	18.974	2.66	0.72
25 °C	27.124	3.00	0.81
35 °C	37.641	3.35	0.90

**Table 3 materials-16-00350-t003:** Kinetic parameters of the PAA/PEI adsorption on the NE–AC activated carbon.

Calculated Parameters	Pseudo-First Order Model			Pseudo-Second Order Model		
	*q_e_* [mg/g]	*k*_1_ [1/min]	*R* ^2^	*q_e_* [mg/g]	*k*_2_ [g/(mg·min)]	*R* ^2^
	PAA					
15 °C	1.02881	8.65444	0.8313	94.3396	0.00185	0.9982
25 °C	1.03624	7.91933	0.9420	102.041	0.00221	0.9999
35 °C	1.03241	7.22797	0.9190	71.9424	0.00472	0.9854
	PEI					
15 °C	1.02316	5.31486	0.9339	30.1205	0.00587	0.9989
25 °C	1.01329	1.22266	0.6614	20.5339	0.02061	0.9995
35 °C	1.02439	6.2257	0.8792	24.2718	0.00362	0.9949

**Table 4 materials-16-00350-t004:** Percentage desorption of polymers from activated carbons surface from the single and mixed adsorbate systems under different temperature conditions.

Desorption [%]			
Desorption Agent	H_2_O	HNO_3_	NaOH
PAA, 15 °C			
NE_AC + PAA	2.44	2.93	61.23
NE_AC + PAA + PEI	3.35	4.70	50.60
SA_AC + PAA	3.86	3.84	57.58
SA_AC + PAA + PEI	4.53	5.56	55.55
PEI, 15 °C			
NE_AC + PEI	13.97	-	-
NE_AC + PEI + PAA	2.79	-	-
SA_AC + PEI	14.81	-	-
SA_AC + PEI + PAA	0.56	-	-
PAA, 25 °C			
NE_AC + PAA	1.36	2.76	61.01
NE_AC + PAA + PEI	1.23	1.81	47.68
SA_AC + PAA	2.53	2.00	46.45
SA_AC + PAA + PEI	1.36	2.42	48.59
PEI, 25 °C			
NE_AC + PEI	44.15	-	-
NE_AC + PEI + PAA	4.52	-	-
SA_AC + PEI	24.73	-	-
SA_AC + PEI + PAA	9.69	-	-
PAA, 35 °C			
NE_AC + PAA	1.55	2.90	56.65
NE_AC + PAA + PEI	2.93	8.85	45.50
SA_AC + PAA	3.91	5.34	54.10
SA_AC + PAA + PEI	5.86	8.32	53.35
PEI, 35 °C			
NE_AC + PEI	36.28	-	-
NE_AC + PEI + PAA	5.40	-	-
SA_AC + PEI	39.29	-	-
SA_AC + PEI + PAA	4.61	-	-

**Table 5 materials-16-00350-t005:** Thermodynamic parameters of PAA and PEI adsorption on activated carbons surface.

Adsorbent	*T* (°C)	*K_c_* (dm^3^/g)	Δ*H*° (kJ/mol)	Δ*S*° (kJ/(mol·K))	Δ*G*° (kJ/mol)
		PAA			
NE_AC	15	5.58	55.4	212.4	−16.8
	25	105.38			−24.7
	35	23.84			−21.7
SA_AC	15	2.17	1.1	10.3	−14.5
	25	2.22			−15.1
	35	2.24			−15.6
		PEI			
NE_AC	15	0.37	5.7		10.9
	25	0.32			
	35	0.43			
SA_AC	15	0.29	11.8		30.9
	25	0.35			
	35	0.40			

## Data Availability

Data are contained within the article.

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
