# Peer review of "Temperature Effect on Ionic Polymers Removal from Aqueous Solutions Using Activated Carbons Obtained from Biomass"

_materials, 2022, doi:10.3390/ma16010350_

Round 1

Reviewer 1 Report

The manuscript entitled “Temperature effect on ionic polymers removal from aqueous solutions using activated carbons obtained from biomass” by Gęca et al. investigates the effect to temperature on the adsorption-desorption characteristics of poly(acrylic acid) (PAA) and polyethyleneimine (PEI) on chemically activated carbon (AC) derived from natural herbs. The changes in the polymer-solvent interaction were monitored by measuring the characteristic length scales (Rh and RMS chain end-to-end distance). Desorption of PAA and PEI was carried out with moderate success using NaOH and water, achieving a maximum desorption of 61% and 44%, respectively. Spectroscopic studies confirmed the presence of stable films on the adsorbent near the adsorbent-adsorbate interface. The work performed is original and is of considerable interest to the broader scientific community. However, there are some minor issues and one major issue (Point 7) associated with the manuscript, as outlined below, that need to be addressed before it can be considered for publication in the Materials journal.

Comments to the Authors:

1.     The Introduction is missing some pertinent references that discuss environmental remediation using chemically modified AC derived from biomass feedstocks. It is recommended that the authors find and cite such articles in the manuscript to provide a background of the current state-of-the-art in the field.

2.     Why did the authors choose the chemical activation method for AC preparation? Control studies using a physical activation method is warranted to differentiate between the two procedures.

3.     Please be careful about the number of significant figures included in Tables 1 & 2.

4.     Section 2.3: How was the AC loading optimized? Did the authors look at different loadings and chose the one that exhibited the best adsorption characteristics? Please comment.

5.     Do the solid lines (linear fits to the data) in Figures 1 and 2 have any physical significance? If not, please consider removing the lines. Also, it might be good to combine Figures 1 and 2 into a single figure.

6.     Section 3.2: Please consider providing the percentage removal data in addition to the adsorption capacity. Figures 1-4, please make sure that the legends have the “°” sign for all the temperatures.

7.     The major limitation of the submitted manuscript is that no attempt was made to monitor the adsorption kinetics (adsorption capacity vs. time). This reviewer feels that the manuscript will significantly improve with the addition of such experiments along with modeling of the adsorption kinetics and isotherms that will present a complete understanding of the mechanism of the process. Further, the authors performed tests at three different temperatures and should try to calculate the relevant thermodynamic parameters (free energy of adsorption).  

Author Response

Reviewer 1

  1. The Introduction is missing some pertinent references that discuss environmental remediation using chemically modified AC derived from biomass feedstocks. It is recommended that the authors find and cite such articles in the manuscript to provide a background of the current state-of-the-art in the field.

The references discuss environmental remediation using chemically modified AC derived from biomass feedstocks were added in the first paragraph of Introduction.

  1. Why did the authors choose the chemical activation method for AC preparation? Control studies using a physical activation method is warranted to differentiate between the two procedures.

Chemical activation was chosen on the basis of our previous research on obtaining activated carbons from plant biomass, for example https://doi.org/10.3390/molecules27238597. This method is much more effective in developing the specific surface and porous structure, as a result of which the obtained carbon materials show much better sorption capacity. However, we would like to explain that the presented research is only a part of the project carried out by the our PhD student and next year it is planned to obtain a series of activated carbons also by physical activation.

  1. Please be careful about the number of significant figures included in Tables 1 & 2.

The number of significant figures included in Tables 1 & 2 was checked. For each parameter the same number of decimal places was used.

  1. Section 2.3: How was the AC loading optimized? Did the authors look at different loadings and chose the one that exhibited the best adsorption characteristics? Please comment.

Before starting adsorption measurements, we always optimize the weight of the solid. We tested various masses of the adsorbent and choose the one that ensures the achievement of adsorption equilibrium in the examined system.

  1. Do the solid lines (linear fits to the data) in Figures 1 and 2 have any physical significance? If not, please consider removing the lines. Also, it might be good to combine Figures 1 and 2 into a single figure.

The linear fits of viscosity data is necessary to determine the intrinsic viscosity [h], from dependence of reduced viscosity (hr) versus the polymer concentration (c) by extrapolating the straight line to c=0. The intrinsic viscosity is next used to the calculations of linear dimensions of polymeric chains in the solution.

  1. Section 3.2: Please consider providing the percentage removal data in addition to the adsorption capacity. Figures 1-4, please make sure that the legends have the “°” sign for all the temperatures.

The percentage removal data were provided and the “°” sign for all the temperatures were added in all Figures.

  1. The major limitation of the submitted manuscript is that no attempt was made to monitor the adsorption kinetics (adsorption capacity vs. time). This reviewer feels that the manuscript will significantly improve with the addition of such experiments along with modeling of the adsorption kinetics and isotherms that will present a complete understanding of the mechanism of the process. Further, the authors performed tests at three different temperatures and should try to calculate the relevant thermodynamic parameters (free energy of adsorption).

The PAA and PEI adsorption kinetics were studied at pH 3 at 15, 25 and 35 °C for NE_AC sample (the time intervals were: 10, 30, 60, 90, 120, 180 min). The obtained data were fitted to the pseudo-first-order and the pseudo-second-order models of adsorption. The results of these studies were presented in Fig. 6 and Table 3.

The modeling of polymers adsorption with the use of classical adsorption models (i.e. Freundlich, Langmuir) is impossible due to the fact that adsorption of huge polymeric chains does not meet the assumptions of these models (single macromolecule occupies many adsorption sites and the multilayer are usually formed).

The free energy of adsorption (∆G°) was calculated using Eqs 5 and 6, whereas entropy and enthalpy were evaluated from the van’t Hoff plot presented in Figure 7. The thermodynamic parameters are presented in Table 5 and discussed in the text.

Reviewer 2 Report

This manuscript focus on the temperature influence on adsorption mechanisms of anionic poly(acrylic acid) and cationic polyethylenimine on the surface of nettle and sage herbs derived activated carbons, which is of interesting to broad readers. After carefully reading, it was found that this article needs major revisions because several issues and explanations are still need to be clarified.

1.      A lot of porous carbon materials have been developed from biomass for waste water treatment. More references are suggested to be cited in the introduction for broad readers, e.g. Journal of Bioresources and Bioproducts 2021, 6 (4), 292-322; Journal of Bioresources and Bioproducts 2022, 7 (2), 109-115; Inorganic Chemistry Frontiers 2022, 9, 6108-6123.

2.      “At pH 3 PEI is almost completely dissociated.” in line 94 needs to be revised.

3.      “15C”, “25C” and “35C” in Figure 1-4 should be revised as “15 oC”, “25 oC” and “35 oC”.

4.      Please add the specific surface area and pore size distribution data since they are important parameters for porous carbon materials. Please refer and cite Biochar 2022, 4 (1), 50.

5.      Please check the writing of superscripts and subscripts.

6.      How about the absorption capacity of anionic poly(acrylic acid) and cationic polyethylenimine in the presence of other ionic polymers?

7.      How about the absorption capacity of ionic polymers compared to other porous carbon materials?

8.      Could the porous carbon be recycled and reused? How about the absorption capacity at different cycle?

9.      Please pay attention to the writing of references. Some journal names are written in full name while most are written in abbreviation.

10.   Most of the references are too old. More references published recently on the synthesis of porous carbon materials and their application in waste water treatment are suggested to be cited.

Author Response

Reviewer 2

  1. A lot of porous carbon materials have been developed from biomass for waste water treatment. More references are suggested to be cited in the introduction for broad readers, e.g. Journal of Bioresources and Bioproducts 2021, 6 (4), 292-322; Journal of Bioresources and Bioproducts 2022, 7 (2), 109-115; Inorganic Chemistry Frontiers 2022, 9, 6108-6123.

The suggested references were cited. Some others manuscript about biomass usage were also added in the first paragraph of Introduction.

  1. “At pH 3 PEI is almost completely dissociated.” in line 94 needs to be revised.

This sentence was revised.

  1. “15C”, “25C” and “35C” in Figure 1-4 should be revised as “15 oC”, “25 oC” and “35 oC”.

The Celsius degrees have been added in the figures.

  1. Please add the specific surface area and pore size distribution data since they are important parameters for porous carbon materials. Please refer and cite Biochar 2022, 4 (1), 50.

We would like to explain that the presented results are a continuation of our previous study, therefore the textural parameters and acidic-basic character of the activated carbons were not discussed in detail in this paper. In the current manuscript, these data are only briefly presented in Table 1. A detail discussion of these parameters has been provided in the paper https://doi.org/10.3390/molecules27217557, as was indicated in the section 2.1.

The indicated reference was cited in another part of the manuscript - in the discussion of XPS results.

  1. Please check the writing of superscripts and subscripts.

The writing of superscripts and subscripts was checked and corrected.

  1. How about the absorption capacity of anionic poly(acrylic acid) and cationic polyethylenimine in the presence of other ionic polymers?

Our research includes, in the first stage, the determination of the mechanism of PAA and PEI interactions with the surface of activated carbons. In the next stages, we consider the use of other ionic and non-ionic polymers, such as polyacrylamide, poly(vinyl alcohol) or poly(ethylene glycol). In the scientific literature, there are no data available on the adsorption of polymers from multi-component solutions on the surface of activated carbons. Therefore, our research can be helpful in elaborating this issue.

  1. How about the absorption capacity of ionic polymers compared to other porous carbon materials?

The maximum amount adsorbed on the NE_AC surface at pH 3 was 273 mg/g for PAA and 156 mg/g for PEI, respectively. The poly(acrylic acid) adsorbed amount on NE_AC activated carbon is greater than on the activated carbon obtained from the cherry stones – 25 mg/g [1] and on the biocarbon obtained from the peanut shells – 50 mg/g, as well as on the carbon adsorbent prepared from corncobs – 80 mg/g [2].

We only have the results of our research, because the review of the literature [3] revealed that other research groups have not dealt with the adsorption of polymers on the surface of porous carbon materials so far.

  1. Wiśniewska, M., Nowicki, P., Nosal-Wiercińska, A., Pietrzak, R., Szewczuk‐Karpisz, K., Ostolska, I., Sternik, D. Adsorption of poly(acrylic acid) on the surface of microporous activated carbon obtained from cherry stones. Colloids and Surfaces A: Physicochemical and Engineering Aspects, 2017, 514, p. 137–145. doi:10.1016/j.colsurfa.2016.11.053.
  2. Wiśniewska, M., Nowicki, P. Simultaneous removal of lead(II) ions and poly(acrylic acid) macromolecules from liquid phase using of biocarbons obtained from corncob and peanut shell precursors. Journal of Molecular Liquids, 2019, 296, 111806. doi:10.1016/j.molliq.2019.111806.
  3. Gęca, M., Wiśniewska, M., Nowicki P., Biochars and activated carbons as adsorbents of inorganic and organic compounds from multicomponent systems – A review. Advances in Colloid and Interface Science, 2022, 305, 102687. doi:11016/j.cis.2022.102687.
  4. Could the porous carbon be recycled and reused? How about the absorption capacity at different cycle?

The activated carbon regeneration process was tested by single desorption using appropriate desorbing agents. Cyclic desorption measurements will be made in our future studies. Thank you for this valuable suggestion.

  1. Please pay attention to the writing of references. Some journal names are written in full name while most are written in abbreviation.

The style of references were corrected.

  1. Most of the references are too old. More references published recently on the synthesis of porous carbon materials and their application in waste water treatment are suggested to be cited.

Some newest references were added. However most of the old references contain information about using procedures and equations, in this case the citation of original manuscripts have been left unchanged.

Reviewer 3 Report

The main aim of this work was to determine influence of temperature on anionic poly and cationic polyethylenimine adsorption mechanism on the surface of activated carbons obtained from the nettle and the sage herbs. This work can be accepted after the completion of the following modifications. 

1. Funding: A research study needs funding, why did the authors suggest that the work was not funded? 

2. Among the many factors, such as temperature, contact time, pH, which has the greatest influence on the adsorption effect. Please give a reason. 

3. It is recommended to compare some industrial adsorption materials to highlight the practical application value of this study. 

4. The references are too old to be representative of the latest research developments. The following literature may be helpful in improving the quality of the manuscript. (Journal of Hazardous Materials, 2022, 426: 128062. Science of The Total Environment, 2021, 757: 143910.)

 5.  XPS: When the author analyzes XPS, it should be explained in conjunction with the literature.

Author Response

  1. Funding: A research study needs funding, why did the authors suggest that the work was not funded?

We would like to explain that study are carried out as part of the statutory activities of the University. They are not funded by any grant from external sources. Therefore, we cannot provide project number, etc. We did the same in our earlier articles published in MDPI journals.

  1. Among the many factors, such as temperature, contact time, pH, which has the greatest influence on the adsorption effect. Please give a reason.

Among many factors influencing the macromolecular compound adsorption, the most important seems to be solution pH, especially in the case of ionic polymers. In such a situation the pH increase causes changes in the polymeric chains ionization which affects considerably their conformation at the solid-liquid interface determining directly the adsorbed amounts. The pH influence was fully characterized in our previous paper (Gęca, M. Wiśniewska, M.; Nowicki, P. Simultaneous Removal of Polymers with Different Ionic Character from Their Mixed Solutions Using Herb-Based Biochars and Activated Carbons, Molecules. 2022, 27, 7557. doi:10.3390/molecules27217557).

Both activated carbons show acidic character of the surface, so at low pH values they are negatively charged, which favors electrostatically adsorption of the cationic PEI, and in a way limits the anionic PAA adsorption. Nevertheless, in the examined system the adsorbed amounts of PAA are greater than PEI. Poly(acrylic acid) adsorption is mostly caused by chemical interactions and hydrogen bonds creation. Greater PAA adsorbed amounts are mostly caused by its coiled conformation at pH 3, in contrast to PEI developed conformation. The obtained results proved that the greatest PAA adsorbed amount occurs at pH 3. It is related directly to its conformation - at pH 3 poly(acrylic acid) assumes the most coiled conformation due to minimal dissociation of its functional groups. For this reason the adsorption of PAA coils in pores is also possible under such pH conditions. As a result, the PAA adsorbed amount is the greatest at pH 3 (dense packing of adsorption layers). At pH 6 and 9 PAA chains are developed, which limits polymer adsorption in porous structure of the adsorbent. PAA adsorption on negative charged activated carbons surface is possible due to hydrogen bonds formation. The pH value have smaller effect of polyethyleneimine adsorption. It is related to high value of PEI pKb, which is over 9, so at all studied pH values polyethyleneimine assumes developed conformation. Similarly to PAA, polyethyleneimine shows the minimally higher affinity to the solid surface at pH 3. In such a case its adsorption is mostly caused by favorable electrostatic interactions. At pH 3 polyethyleneimine is completely dissociated (it has positive charge) and the acidic activated carbons are negatively charged at this pH value. Thus the polymer molecules undergo attraction by the adsorbent surface.

  1. 3. It is recommended to compare some industrial adsorption materials to highlight the practical application value of this study.

Two different commercially available activated carbon materials were used – purolite (PRT) (Lenntech, Delfgauw, The Netherlands) with specific surface area 900-1000 m2/g and lewatite (LWT) (Lanxess, Dortmund, Germany) characterized by specific surface area 550-650 m2/g. The obtained polymer adsorbed amounts from single and mixed solutions were presented in Fig. 5 and compared with those obtained for examined plant-based activated carbons.

  1. The references are too old to be representative of the latest research developments. The following literature may be helpful in improving the quality of the manuscript. (Journal of Hazardous Materials, 2022, 426: 128062. Science of The Total Environment, 2021, 757: 143910).

The suggested references were cited, some newest references were also added. However most of the old references contain information about using procedures and equations, in this case the citation of original manuscripts have been left unchanged.

  1. XPS: When the author analyzes XPS, it should be explained in conjunction with the literature.

The relevant part of the manuscript has been enriched with references, as suggested by the Reviewer.

Round 2

Reviewer 2 Report

The manuscript has been revised according to the comments and could be accepted now.

Reviewer 3 Report

 Accept in present form